# Predicting the Effect of Mo Addition on Metastable Phase Equilibria and Diffusion Path of Fe in NiAl Laser-Clad Coatings Using First-Principle Calculations and CALPHAD Simulations

Chun-Ming Lin





Department of Mechanical Engineering, Minghsin University of Science and Technology, Hsinchu 30401, Taiwan; chunming@must.edu.tw; Tel.: +886-3-559-3142 (ext. 3013)

**Abstract:** This study used first-principle calculations and CALPHAD simulations to investigate the effects of adding Mo to NiAl laser-clad coatings in terms of metastable phase equilibria and Fe diffusion path with a focus on thermodynamic phase stability and element diffusion behavior. First-principle calculations were performed using $3 \times 3 \times 3$ supercells to determine the formation energies of NiAl and Mo-rich phases within a Mo-doped NiAl cladding layer. The findings of this analysis are consistent with the d-orbital energy and bond order results obtained using DV-Xa molecular orbital calculations and phase diagrams obtained using Thermo-Calc simulations. The results also revealed that the substitution of Ni and Al atoms for Fe and Mo in the NiAl matrix decreased the stability of the $B_2$ structure, thereby reducing phase formation energy. DICTRA simulations were also performed to characterize the diffusion behavior of Fe from the substrate to the surface of the coating. This analysis revealed that the rate of Fe diffusion was slower in the Mo phase than in the NiAl phase. Furthermore, the rate of Fe diffusion in molten material was inversely proportional to the Mo content. These results are consistent with the substitution mechanism used to describe diffusion, wherein diffusivity is inversely proportional to Mo content, due to its high melting point and the fact that un-paired electrons in the outer shell of Mo atoms increase the bonding strength, thereby hindering the diffusion of Fe. Due to the high cooling rates involved in the laser-cladding process, DICTRA simulations tend to overestimate the Fe diffusion distance. Nonetheless, the theoretical results obtained in this study were in good agreement with experiment observations (EPMA line scans). These results confirm the feasibility of using quantum modeling techniques and first-principle calculations to predict the effects of Mo addition on phase formation and element diffusion behavior in the NiAl laser-cladding process.

**Keywords:** Mo-doped NiAl coating; laser deposition; phase stability; computational model; diffusion mechanism

## 1. Introduction

Near-equiatomic intermetallic compounds (IMCs) feature a high melting temperature, low density, and superior anti-corrosion and anti-oxidation properties [1–6]. NiAl IMCs in particular provide high thermal and electrical conductivity with excellent anti-oxidation properties [7–9]; however, low fracture toughness at room temperature severely limits its practical applicability [10,11]. A number of research teams have sought to enhance the mechanical properties of NiAl. Schulson and Barker [12] reported the occurrence of a brittle-to-ductile transition when the grain size was reduced to less than 20 µm, thereby increasing mechanical properties. Microalloying [13–15] or macroalloying [16–19] NiAl with ternary elements has also been shown to improve fracture toughness, particularly through the addition of Mo [20–22]. Advances in laser surface cladding have recently made it possible to combine micro/macroalloying with grain refinement in a single process [23–25]. In a previous study [26], the current authors investigated the inclusion of Mo in the NiAl matrix of laser-clad coatings. In that study, the diffusion of Fe from the substrate resulted

in phase separation of the NiAl with a corresponding improvement in the mechanical properties; however, a number of issues were left unanswered [26]. For example, EPMA results revealed that the Mo-rich phases contained large quantities of Ni and Al, whereas the phase diagrams derived through Thermo-Calc simulations (TCNI8 database) indicated that the Mo-rich phases consisted almost entirely of Mo (note that the model used in the simulations did not account for the mixing of Mo in the NiAl phase). Furthermore, we were unable to elucidate the Fe diffusion path from the substrate to the coating surface during laser processing, thereby undermining our ability to predict the composition of the coating or the characteristics of the resulting materials.

The time and financial costs of experimentation have prompted the use of computational methods based on quantum theory to reveal the mechanisms underlying element diffusion and the thermodynamics of phase formation. One approach to modeling diffusion and phase formation is the DICTRA thermodynamic model, based on the CALPHAD simulations and corresponding thermodynamic database, which includes the chemical, structural, and mechanical properties of numerous multiphase materials [27]. Nonetheless, the derivation of material properties via first-principle calculations generally enables predictions of higher accuracy [28–31]. Computational quantum mechanics models based on Density Functional Theory (DFT) [32–34] have also made it possible to estimate the underlying electron density as well as the optical and thermodynamic properties of materials simply by calculating electron interactions.

In the current study, we sought to fill previous research gaps [26] by performing a detailed theoretical investigation into thermodynamic phase stability and kinetic element diffusion in NiAl laser-clad coatings doped with various quantities of Mo (0~15 wt%). Strong agreement between theoretical results and experiment observations verified the efficacy of the proposed computational modeling as a basis for the design and optimization of laser-clad coatings.

## 2. Experimental Details and Aims

NiAlMo$_x$ composite coatings were prepared with added Mo (0, 3, 6, 9, 12, and 15 wt%) on medium-carbon-steel substrates using a laser-cladding technique. Briefly, a mixture of the constituent materials (in powdered form) was applied to the surface of the substrate and then scanned by a continuous-wave $CO_2$ laser system using power of 1.1 kW, a beam diameter of 4.5 mm, and scanning speed of 400 mm/min. The scanning process resulted in the formation of a cladding layer with a thickness of approximately 1.5 mm. The microstructure of the composite coating was characterized using an electron probe microanalyzer (EPMA, JEOL JXA-8600SX) and X-ray diffraction (XRD, Rigaku, Tokyo, Japan) using Cu K$\alpha$ radiation.

The phase formation and thermodynamic stability of the Mo-doped NiAl composite coatings were derived using first-principle calculations. The total energy of the various phases was determined via simulations using Materials Studio 7.0 (Cambridge Serial Total Package; CASTEP) based on the generalized gradient approximation (GGA) of Perdew–Burke–Euzerhof (PBE) [35]. The simulations considered simple $3 \times 3 \times 3$ supercells of NiAl and Mo-doped NiAl with a BCC structure. For comparison purposes, DV-Xa molecular orbital calculations were performed to determine the d-orbital energy and bond order of the various supercells. We also formulated phase diagrams of the doped NiAl structures via Thermo-Calc simulations (TCNI8 database) [27]. Finally, DICTRA simulations (TCN18 database) were used to investigate the inter-diffusion pathways of four binary systems, including Fe-NiAl$_{(s)}$, Fe-Mo$_{(s)}$, Fe-NiAl$_{(l)}$, and Fe-NiAl-3, 9, or 15 wt% Mo$_{(l)}$. For validation purposes, the first-principle calculation results and DICTRA inter-diffusion results were compared with experimental data and the findings reported in previous theoretical studies.

## 3. Results and Discussion

It was reported in ref. [26] that the addition of Mo to NiAl compounds produces an intermetallic dual-phase matrix consisting of Mo-rich BCC phase and NiAl. However, as

described in Section 1, that study left two key issues unresolved: (1) The EPMA results indicated the presence of large quantities of Ni and Al in the Mo-rich phases, whereas the Thermo-Calc simulations indicated that the Mo-phase was nearly pure; and (2) that study was unable to explain the presence of large quantities of Fe at the surface of the cladding, particularly in the Mo-rich phases. The primary objective in this study was to address these issues using first-principles calculations and DICTRA simulations.

Figure 1 presents an NiAl-Mo equilibrium phase diagram obtained via Thermo-Calc simulations. Overall, the volume fraction of Mo-rich phase increased proportionally with an increase in Mo content. Intermetallic compounds of NiAl with a large electronegativity difference between Ni and Al (i.e., $\Phi$ = 1.91 and 1.61, respectively, $\Delta\Phi$ = 0.3) generally display high bond strength, which tends to restrict the solubility of other impurities, such as Mo. The Hume–Rothery rule regarding the formation of solid solution phases stipulates that the large differences in electronegativity between Mo ($\Phi$ = 2.16) and Ni and Mo and Al inhibit the substitution of Mo for Ni ($\Delta\Phi$ = 0.25) or Al ($\Delta\Phi$ = 0.55) in NiAl compounds, thereby suppressing NiAlMo phase formation. Under these conditions, the phase separation (i.e., spinodal decomposition) of Mo-rich phases in NiAl is inevitable. However, under the high energy levels involved in the laser-cladding process, it is important to consider the diffusion of Fe atoms from the steel substrate in the process of phase formation. Figure 2 presents a phase diagram obtained using Thermo-Calc simulations for the NiAl-xMo (x = 3, 9, 15 wt%) IMC with various molar fractions of Fe. It is clear that the Mu phase and Fe-rich phase formed at low temperatures (500 to 1000 K). The formation of the Mu phase can be attributed to the low solubility of the NiAl IMC, which prompted the Fe to react with the residual Mo-rich phase. Increasing the Mo content was shown to right-shift regions pertaining to the Mu phase and Fe-rich phase under the effects of Fe dilution [26]. In the following, first-principle calculations and molecular orbital analysis were used to evaluate the correctness of phase diagrams generated from Thermo-Calc simulations under the assumption of thermodynamic equilibrium.

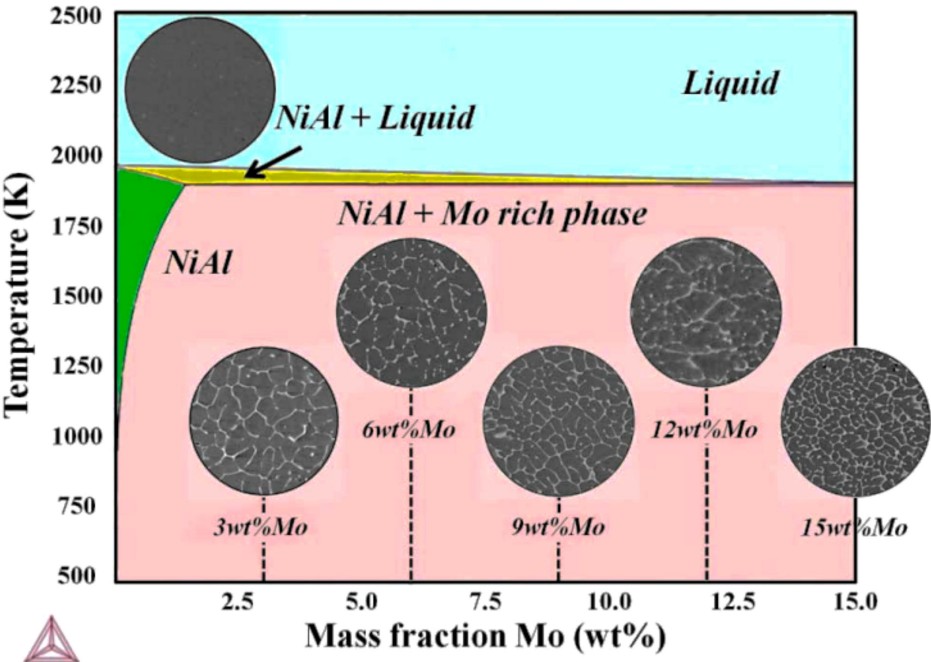

**Figure 1.** Equilibrium phase diagram of NiAl-xMo (x = 3, 9, and 15 wt%).

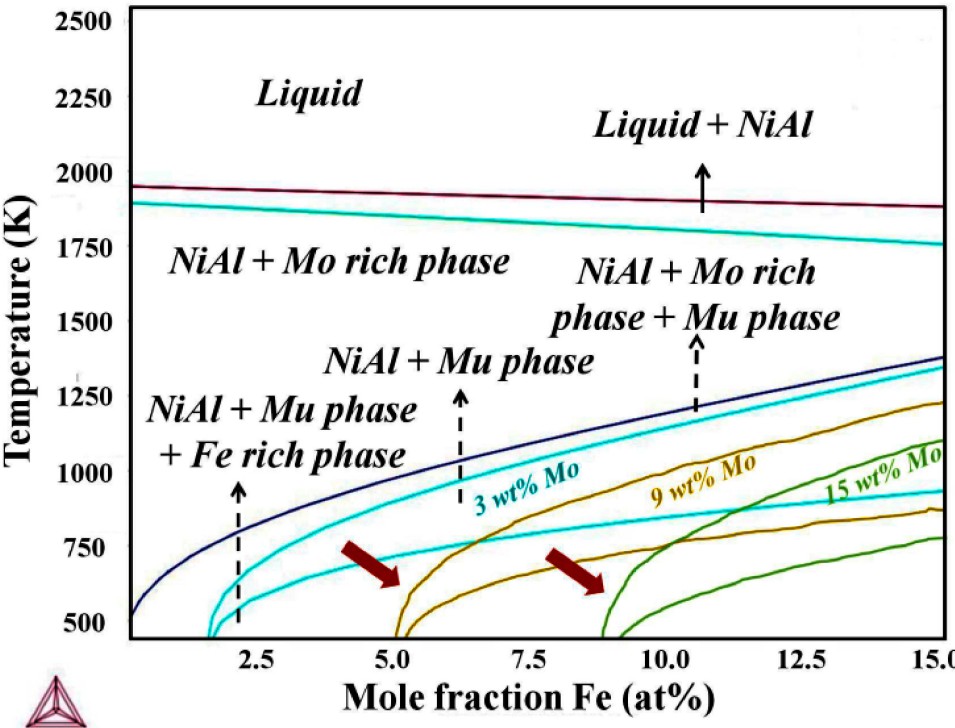

**Figure 2.** Phase diagram of NiAl-xMo (x = 3, 9, and 15 wt%) containing various proportions of Fe.

First-principle simulations based on known crystal structures have been widely used to predict the stability of phases and precipitates as well as the binding preferences among vacancies and impurities [28–31]. The equilibrium state of phases can be obtained by calculating the formation enthalpy ($\Delta H$) of a given component. In the current study, the unresolved issue of Mo-rich phase formation was addressed by calculating the formation enthalpies of various elements in solution with NiAl with the aim of determining whether the crystal structure is truly in an equilibrium state (as assumed in the Thermo-Calc simulations). In general, the equilibrium $\Delta H$ (per atom) can be obtained using the method outlined in ref. [36], as follows.

$$\Delta H = \frac{E\left(Ni_\alpha Al_\beta Fe_\gamma Mo_\delta\right) - \alpha E(Ni) - \beta E(Al) - \gamma E(Fe) - \delta E(Mo)}{\alpha + \beta + \gamma + \delta} \tag{1}$$

where $E\left(Ni_\alpha Al_\beta Fe_\gamma Mo_\delta\right)$, $E(Ni)$, $E(Al)$, $E(Fe)$ and $E(Mo)$ are the total energies of the constructed intermetallic supercells and the individual energies of each of its constituent parts (i.e., Ni, Al, Fe, and Mo in the current study). Figure 3b–d present the supercells used in first-principle calculations constructed in accordance with the XRD patterns in Figure 3a [26]. Table 1 lists the results for the formation enthalpies of the four supercells obtained using first-principle calculations. Figure 3b,c respectively present the supercells of pure NiAl and as-deposited NiAl. The as-deposited NiAl matrix contained a large quantity of Fe (i.e., diffused from the substrate), which was not in the pure NiAl. First-principle calculations revealed that the crystal structure of the as-deposited NiAl coating involved the substitution of 2 Fe atoms at each Ni and Al site in the NiAl matrix (i.e., NiAl (2Fe)). As shown in Table 1, the change from pure NiAl to Fe-substituted NiAl increased the $\Delta H$ of the entire system. In other words, Fe substitution resulted in a compound that was more stable than the NiAl. Figure 3d,e respectively present supercells of as-deposited NiAl coatings with added Mo at various concentrations (3–9 wt% and 12–15 wt%). The crystal structures of the two corresponding NiAl coatings respectively involved the substitution of 2 Mo and 2 Fe atoms (2Mo2Fe), and 3 Mo and 1 Fe atoms (3Mo1Fe) at each Ni and Al site. In other words, increasing the amount of added Mo increased the number of Mo atomic substitutions at Ni and Al sites. As shown in Table 1, Mo substitution increased

the formation enthalpy of the entire system. Moreover, the formation enthalpy of the Mo-doped NiAl was more pronounced than that of the Fe-substituted NiAl. The formation energies of the various phases were ranked as follows: NiAl < NiAlFe < NiAlFeMo < NiAlMo. In other words, Mo substitution decreases the stability of the NiAl structure. The fact that Mo is less likely than Fe to form a solution in NiAl means that pure Mo phase can readily form within the NiAl matrix. In terms of NiAl and Mo-rich phase separation, our first-principle results are consistent with those obtained using Thermo-Calc simulations.

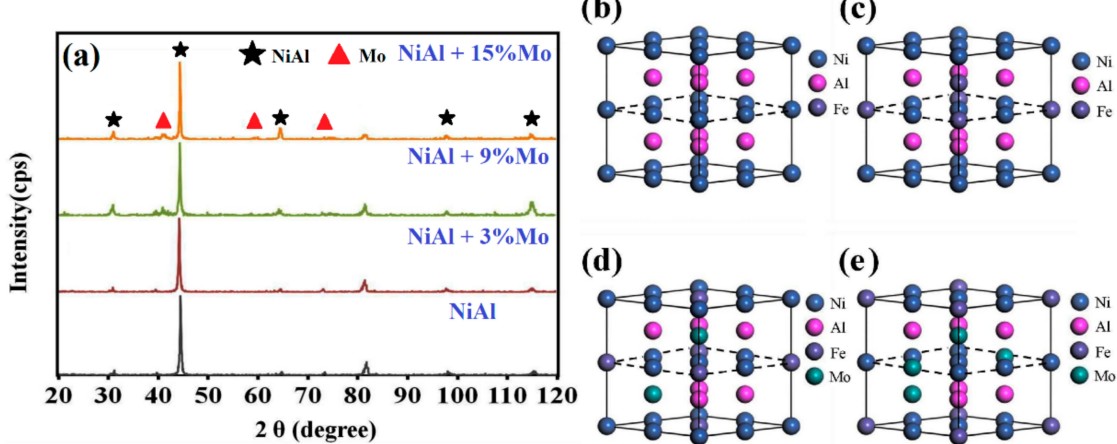

**Figure 3.** (**a**) XRD patterns of pure NiAl and NiAl-xMo (x = 3, 9 and 15 wt%); and supercells used in first-principle calculations for (**b**) NiAl; (**c**) NiAl (2Fe); (**d**) NiAl (2Mo2Fe); and (**e**) NiAl (3Mo1Fe).

**Table 1.** First-principle calculation results for the formation enthalpy (KJ/mol) of supercells shown in Figure 3.

| Supercell | Ni | Al | Fe | Mo | Structure | Formation Enthalpy |
|---|---|---|---|---|---|---|
| NiAl | 8 | 8 | - | - | Pm$\bar{3}$m | −65.34 |
| NiAl (2Fe) | 7 | 7 | 2 | | Pm$\bar{3}$m | −58.89 |
| NiAl (2Mo2Fe) | 6 | 6 | 2 | 2 | Pm$\bar{3}$m | −46.08 |
| NiAl (3Mo1Fe) | 6 | 6 | 1 | 3 | Pm$\bar{3}$m | −34.96 |
| Mo | - | - | - | 16 | Pm$\bar{3}$m | −1936.89 |

As mentioned above, all systems were disordered at high temperatures but formed an ordered structure as the system was cooled. Figures 1 and 2 illustrate the formation of structures in an MoNiAl alloy system as a function of Mo content. Lowering the temperature of the system to below the ordering temperature, Tc, resulted in the strong ordering of at least two of the $n > 4$ TM elements, but not all of them. Cooling resulted in a system with Ni, Fe, and Mo largely segregated to one sublattice with Al segregated to the other. In simulations aimed at quantitatively reproducing the effects of heating and cooling, the system was essentially in equilibrium. This study also examined the effects of Mo content by comparing NiAl, NiAlFe, NiAlFeMo, and NiAlMo alloy systems. In the NiAlFe alloy with higher Mo content, Tc remained above the melting point, such that the solid phase of the alloy never become disordered, i.e., the alloy maintained partial ordering at the melting temperature. In the NiAlFe with lower Mo content, Tc occurred in the solid phase at 700–750 °C. Note that a more complete description of trends in B2-ordering is presented below. The low fraction of Al–Al pairs at all temperatures indicated that even at temperatures above the B2-transformation temperature, the Al atoms paired primarily with TMs. Interestingly, there was a decrease in the fraction of Al–Al pairs corresponding to an increase in Mo content, which is the opposite of what would normally be expected for a disordered material. Thus, it appears that even in a disordered BCC phase

at high temperature, the Al atoms presented short-range order, such as that associated with miscibility gaps (i.e., a strong tendency toward ordering). Note that this contradicts previous descriptions in which Mo is assumed to be randomly organized.

The first-principle results in this study were also consistent with those obtained using the molecular orbital method, which is based on the d-orbital energy levels (Md) and bond order (Bo) electronic parameters [37]. DV-Xa molecular orbital calculations have previously been used to predict the phase stability and mechanical properties of various alloys [37–39]. Md is associated with the electronegativity and atomic radius of the elements used in alloying as well as the ionic interaction among the atoms. By contrast, Bo is associated with the overlapping outer electrons (d or p) of two atoms, as measured by the strength of their covalent bonds. Differences in the electronegativity of the transition metals in the current study (Fe ($\Phi$ = 1.83); Ni ($\Phi$ = 1.91); Mo ($\Phi$ = 2.16)) correspond to the number of un-filled electrons, which tend to interact with Al electrons filling the *p*-orbital. The fact that this can induce the formation of covalent bonds and ionic interactions means that Md and Bo could be used to estimate the stability of phases within a given crystal structure, the results of which could be compared with those based on first-principle calculations. Thus, the intermetallic compounds of MAl [38] were then used to calculate the Md and Bo values of $(Ni_xFe_yMo_z)Al$ in order to simulate the structures illustrated in Figure 3 For each structure, the d-orbital energy level and bond order were respectively computed as follows:

$$Md = x(Md)_{Ni} + y(Md)_{Fe} + z(Md)_{Mo}, \tag{2}$$

$$Bo = x(Bo)_{Ni} + y(Bo)_{Fe} + z(Bo)_{Mo}, \tag{3}$$

where $(Md)_{Ni}$, $(Md)_{Fe}$, and $(Md)_{Mo}$ respectively refer to the d-orbital energy levels of *Ni*, *Fe*, and *Mo* in the $M_2Al_4$ cluster, while $(Bo)_{Ni}$, $(Bo)_{Fe}$, and $(Bo)_{Mo}$ indicate the corresponding bond order.

Figure 4a compares first-principle calculations and molecular orbital calculations in terms of formation enthalpy, d-orbital energy level (Md), and bond order (Bo) in the four supercells in Figure 3. Formation enthalpy increased with an increase in Mo content, which resulted in a decrease in Fe content in the NiAl IMC. The positive correlation between the quantities of Mo/Fe and Md can be attributed to the large atomic radius and low electronegativity of Mo and Fe. The increase in Bo with an increase in Mo can be attributed to the larger number of un-filled Mo and Fe electrons, resulting in a high covalent strength and strong ionic interactions. The intermetallic MAl structure map [38] in Figure 4b shows that all of the results of first-principle calculations fell within the area of the $B_2$ structure; however, increasing the content of Mo or Fe in the NiAl led to the transformation of other structures [40,41], with a corresponding increase in the electronic density of the 3-d elements. For example, additional electrons in the NiAl alloy were shown to occupy a corner of the first Brillouin zone, leading to the transformation of the Fermi surface, which is unique to each compound. The availability of additional elements in the NiAl IMC increased the number of electrons and in so doing increased the probability of electron interaction, with a corresponding increase in the Fermi surface volume under the effects of crystal structure alteration [42]. The superfluous electrons supplied by Mo and Fe further promoted phase transformation, thereby decreasing the stability of the $B_2$ phase and increasing the formation enthalpy (as indicated in first-principle calculations).

It is possible to investigate the mechanisms underlying phase diagrams at the atomic scale by combining first-principle calculations with DV-Xa molecular orbital calculations. In other words, this approach to modeling can be used to explain thermodynamic phenomena in solids and predict the formation of phases in the design of alloys. Note that the first-principle calculations verified the correctness of the Thermo-Calc phase diagrams; however, this does not explain the large quantities of Ni and Al in the Mo-rich phases. In many as-cast alloys, it is impossible to attain thermodynamic equilibrium, particularly under the solidification conditions associated with laser cladding [26,43]. Mixing the NiAl and Mo powders prior to laser cladding limits the diffusion distance between the various elements

to the μm to nm scale (equal to radius of particles in the powder). With the short diffusion distance and high solidification rate of laser cladding, there is insufficient time to attain thermodynamic equilibrium. Thus, even though the phase diagrams describe a separation of NiAl and Mo-rich phases, it appears that the Ni and Al may actually remain trapped within the Mo-rich phase.

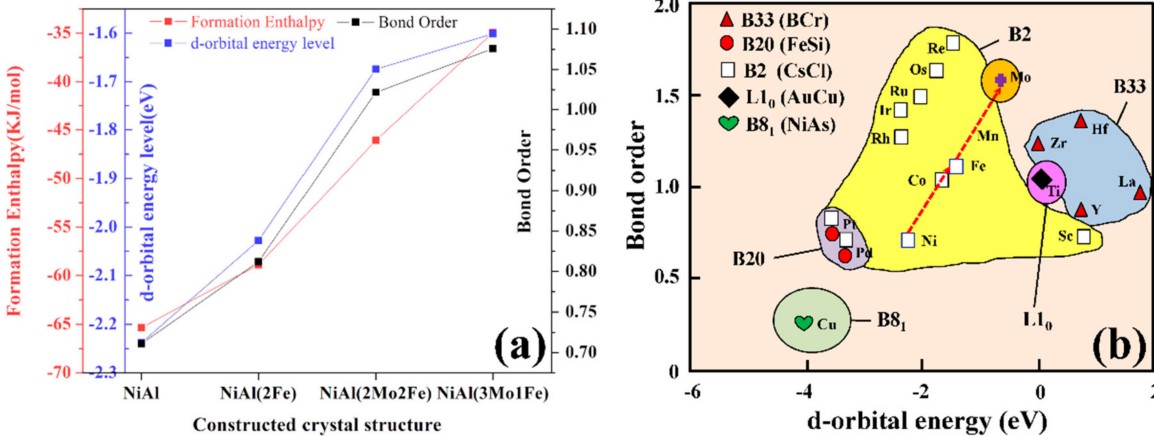

**Figure 4.** (**a**) Formation enthalpy, d−orbital energy, and bond order of the various supercells; (**b**) structure map showing the substitution of various elements in MAl intermetallic compounds [38].

　　　The high scanning speed of the laser (400 mm/min) results in a very short contact time (i.e., 0.5 s to 1 s), which in turn reduces the time available for the diffusion of Fe from the substrate into the coating. The rapid solidification of the coating (104~106 °C/s) further restricts the diffusion of Fe. Nonetheless, in the previous study [26], EPMA results revealed large quantities of Fe at the coating surface, which is inconsistent with previously observations. This raises the question of how the Fe is able to diffuse over such long distances. In the laser-cladding process, the transfer of laser energy occurs under two conditions: (1) Initial transfer of energy into the mixed powder on the substrate at the beginning of the cladding process; and (2) subsequent energy transfer associated with the 50% overlap in each successive pass of the laser. Figure 5 presents DICTRA simulation results indicating the Fe diffusion path from the substrate into the coating layer under the following conditions: (1) A solid state NiAl matrix (see Figure 5a) with Mo-rich phases (see Figure 5b); and (2) a liquid-state coating with various proportions of Mo (see Figure 5c). The results in Figure 5a,b show that Fe diffused only a very short distance in the case with a solid-state NiAl matrix and Mo-rich phases. The diffusivity of Fe in the NiAl matrix far exceeded that in the Mo-rich phases. In other words, the presence of Mo-rich phases slowed the diffusion of Fe through the solid-state material. Increasing the Mo content eventually resulted in an even distribution of Mo-rich phases in the coating, thereby decreasing the Fe diffusion distance even further. It is for this reason that the results in Figure 5a,b fail to explain the existence of Fe at the coating surface. From this, one may infer that most of the observed Fe diffusion would not occur in a solid-state system. Figure 5c presents the Fe diffusion paths in the liquid-state coating at 2500 K at 0.5 s and 1 s. At first glance, it appears that the Fe diffusion paths in the pure NiAl coating and the sample with added Mo are similar; however, under close inspection, it was observed that the diffusion of Fe into the coating was inversely proportional to Mo content. A comparison of results in Figure 5a,b revealed that some Fe managed to diffuse to the edge of the coating, which resulted in the accumulation of Fe at coating surface (as previously detected in the EPMA results [26]). Overall, the results in Figure 5 suggest that most of the Fe diffusion occurred in a liquid state (initial deposition). The rapid solidification associated with laser cladding limits the time available for Fe diffusion in a liquid state, due to the fact that the melting pool freezes almost immediately. Nonetheless, the high diffusivity of Fe in a liquid state permits the rapid diffusion of Fe to the surface of the melting pool. Even after the coating has solidified, residual

laser energy enables the diffusion of Fe to continue through the solid coating. In this situation, the low diffusivity of Fe in a solid state limits the Fe diffusion distance. Thus, the DICTRA simulation results suggest that in the laser-cladding process, Fe diffusion occurred in a liquid state as well as solid state at a ratio of approximately 3:1. Accordingly, Fe diffusivity can be calculated using the methods outlined in [44,45], as follows:

$$D = a^2 v exp\left(\frac{-\Delta G_v}{RT}\right)exp\left(\frac{-\Delta G_m}{RT}\right)$$ (4)

where $a$, $v$, $\Delta G_v$, and $\Delta G_m$ respectively indicate the lattice constant, Debye frequency ($\cong 10^{12} \sim 10^{13}$ s$^{-1}$), vacancy formation free energy, and migration activation energy. In the DICTRA simulations, the diffusion of all the Mo-doped NiAl IMCs occurred at the same temperature. In other words, $a$, $v$, $R$, and $T$ can be regarded as constants and Equation (4) simplified as follows:

$$D \propto exp(-\Delta G_v)exp(-\Delta G_m)$$ (5)

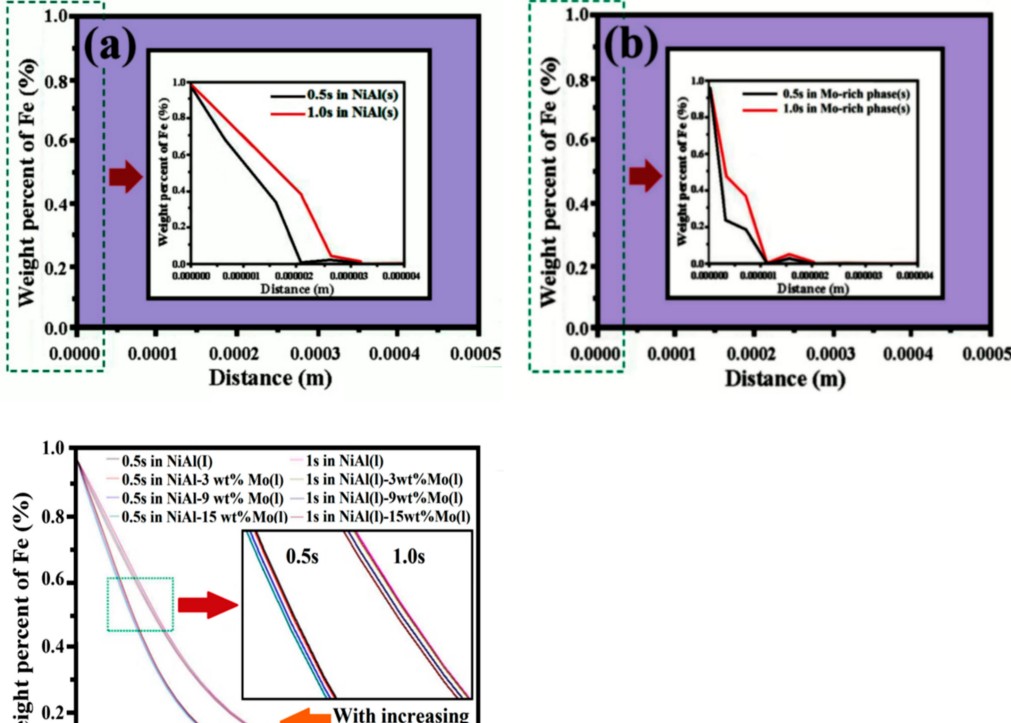

**Figure 5.** DICTRA simulation results for Fe diffusion from substrate to the surface of the coating layer in (**a**) solid-state NiAl; (**b**) solid-state Mo-rich phase at 1500 K; and (**c**) liquid-state coatings with various proportions of Mo at a temperature of 2500 K.

Thus, Fe diffusivity is directly correlated with the $\Delta G_v$ of the pure NiAl intermetallic compound and $\Delta G_m$ of the Fe atoms diffused into the Mo-doped NiAl intermetallic compounds. The values of $\Delta G_v$ and $\Delta G_m$ of Ni, Al and Mo are listed in Table 2. The addition of Mo to the NiAl matrix led to a reduction in the $\Delta G_m$ of Fe and the entire system, which resulted in a lower Fe diffusivity. As shown in Figure 5, Fe diffusion distance was inversely proportional to Mo content, which means that the underlying diffusion mechanism could be described from two perspectives: (1) the high melting point of Mo, and (2) the high Md and Bo values of Mo-doped NiAl IMCs (see Figure 4a). The high melting point of Mo reduces the bonding strength of Mo-doped NiAl IMCs. The resulting increase in $\Delta G_v$ and

$\Delta G_m$ destroys the bonds between the Mo atoms and the Ni and Al atoms in the Mo-doped NiAl IMCs. Meanwhile, the high Md and Bo values associated with the large number of unpaired Fe and Mo atoms increase the strength of the covalent bonds and ionic interactions, thereby hindering the diffusion of Fe.

**Table 2.** Detailed parameters of vacancy formation free energy; $\Delta G_v$, migration activation energy of Fe in each element; $\Delta G_m$, and the activation energy of vacancy, $Q$ [45–49].

|        | $\Delta G_v$    | $\Delta G_m$    | $Q$             |
|--------|-----------------|-----------------|-----------------|
| **Ni** | 1.51 eV [48]    | 1.13 eV [49]    | 2.98 eV [48]    |
| **Al** | 0.68 eV [48]    | 1.36 eV [45]    | 1.33 eV [48]    |
| **Mo** | 2.98 eV [47]    | -               | 4.22 eV [46]    |

Thus, based on Equation (5) and Table 2, it can be inferred that the diffusion of Fe in the Mo-doped NiAl IMCs is somewhat sluggish. Note that this inference is consistent with the results of DICTRA simulation in Figure 6. Figure 6a presents a schematic illustration showing the laser-cladding process performed in the current study. Figure 6b,c respectively present the diffusion paths of elemental Fe, Ni, Al, and Mo in the melting pool of the as-deposited NiAl coating and as-deposited NiAlMo$_x$ coating. Figure 6d,e present the corresponding diffusion conditions and phase formations at the atomic scale. Note that the arrows in Figure 6b–e indicate the direction of diffusion, where the white arrows indicate the diffusion of Ni, Al, and Mo from the outside toward the interior, whereas the black arrows indicate diffusion from the substrate outward. Figure 6d,e present four characteristic areas, labeled I~IV. Area I is adjacent to the laser heat source. The red circles in this area indicate the Fe diffusion path within time interval t = 0~0.5 s, during which the Fe atoms diffuse rapidly through the melting pool (as shown in the DICTRA results). As shown in Areas I to IV, the rate of diffusion was inversely proportional to the temperature of the system, as indicated by the length of the arrows. Figure 6d shows that as the laser passed from the left to the right side of the powder bed, the pure NiAl on the right side combined with the Fe substrate, resulting in the formation of NiAl-Fe on the left side via a process of Fe substitution. Figure 6e shows a similar substitution of Fe in the NiAl and Mo phases, as well as the inter-diffusion of Ni, Al, and Mo in each phase.

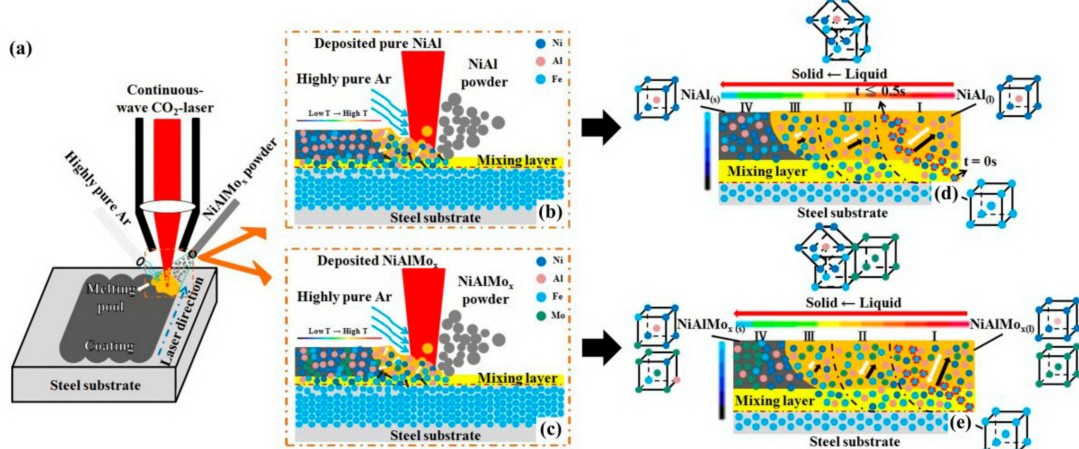

**Figure 6.** (**a**) Schematic illustration of laser-cladding process; and conditions affecting element migration in (**b**) as-deposited NiAl coating; and (**c**) as-deposited NiAlMo$_x$ coating corresponding to diffusion and phase conditions shown in (**d**,**e**).

The validity of the DICTRA simulations and theoretical evaluations was confirmed via comparison with experiment-derived EPMA line scans obtained from cross-sections

that included the substrate and coating. Figure 7 presents the EPMA line scans from a coating of pure NiAl (see Figure 7a) and NiAl coatings containing Mo at concentrations of 3, 9, and 15 wt% (see Figure 7b–d, respectively). Figure 7a shows the inter-diffusion zone with a sharp composition gradient between the substrate and the coating (as indicated by the two vertical dashed lines). The DICTRA simulation results are shown in Figure 5a.

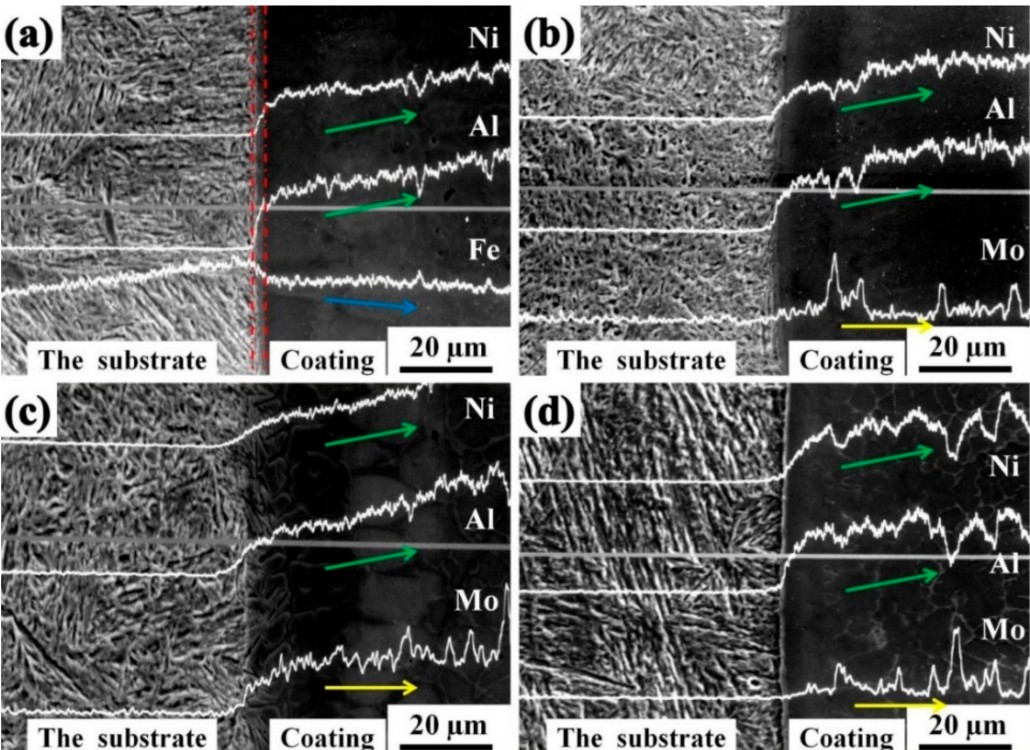

**Figure 7.** EPMA line scan results for (**a**) pure NiAl; and Mo-doped NiAl with (**b**) 3 wt% Mo; (**c**) 9 wt% Mo; and (**d**) 15 wt% Mo.

The results indicate that Fe diffusion in the inter-diffusion zone occurred under solid-state conditions. Ni and Al content were proportional to the distance from the interface, whereas the Fe content was inversely proportional (as indicated by the arrows in Figure 7a). Detailed analysis revealed an increase in Mo content without a significant increase in Ni or Al content (see Figure 7b–d), thereby obstructing the diffusion of Fe. This observed phenomena could be explained by the high cooling rate of the laser-cladding process, which does not provide sufficient time for Fe to diffuse through the liquid coating. As IMCs were deposited on the substrate, the liquid-state coating transformed almost immediately into a solid, whereupon the solid-state diffusion of Fe resulted in the formation of an inter-diffusion zone. Nonetheless, higher laser energy closer to the source of the laser (i.e., above the interface) maintained the coating in a liquid state, such that only a small proportion of the Fe accumulated in the inter-diffusion zone continued its inward diffusion.

## 4. Conclusions

This study used first-principle calculations to characterize the formation enthalpy of pure NiAl IMCs and various Mo-rich phases in Mo-doped NiAl coatings prepared via laser cladding. The calculation results confirmed the existence of a dual-phase NiAl matrix with a Mo-rich phase, and are consistent with the phase diagrams obtained via Thermo-Calc simulations. The relatively high formation energies of the as-deposited NiAl and Mo-rich phases can be attributed to a large difference between the solute atoms and NiAl matrix in terms of electronegativity. The results obtained from DV-Xa molecular orbital calculations confirmed that the stability of these structures was low. These results also revealed that the substitution of Fe (from the substrate) and Mo atoms in the NiAl matrix induced a high



d-orbital energy level (Md) and (bond order) Bo throughout the entire system, resulting in strong ionic interactions and high-strength covalent bonds. In other words, the instability of the present $B_2$ supercells can be attributed to the strong electronegativity of the 3d atoms, which possess numerous un-paired electrons and a correspondlingly high-energy state in the Brillion zone. DICTRA simulation results revealed that the rate of Fe diffusion was slower in the Mo phase than in NiAl phase. Furthermore, the rate of Fe diffusion in the liquid state was inversely proportional to Mo content. These results can be attributed to the high melting point of Mo and the large number of un-paired electrons, which reduced the diffusivity of Fe throughout the entire system. Diffusion calculations revealed that the addition of Mo increased the vacancy formation energy and migration activation energy, both of which contributed to an increase in the activation energy of Fe diffusion, thereby reducing Fe diffusivity. In previous research, DICTRA simulation results pertaining to Fe diffusion were not entirely consistent with experimental observations. Specifically, the high cooling rate associated with the laser-cladding process reduced Fe diffusion by limiting the time available for the diffusion of Fe through the liquid coating. As IMCs were deposited on the substrate, the liquid-state coating transformed almost immediately into a solid, whereupon the subsequent solid-state diffusion of Fe resulted in the formation of an inter-diffusion zone. However, higher laser energy close to the source of the laser (i.e., above the interface) maintained the coating in a liquid state, such that only a small quantity of the Fe accumulated in the inter-diffusion zone continued its inward diffusion. To the best of the authors' knowledge, this is the first attempt to combine quantum computational modeling based on first-principle simulations with DV-Xa molecular orbital calculations. The results confirm the feasibility of the proposed modeling approach in predicting the stability of phases that form in laser-cladding coatings.

**Funding:** This research funding was provided by Ministry of Science and Technology of Taiwan for the financial support of this study under Contract No. MOST 105-2218-E-027-011-MY3.

**Institutional Review Board Statement:** Not applicable.

**Informed Consent Statement:** Not applicable.

**Data Availability Statement:** Not applicable.

**Acknowledgments:** The author gratefully acknowledge An-Chou Yeh of National Tsing-Hua University for his support in the use of Thermo-Calc software.

**Conflicts of Interest:** The authors declare no conflict of interest.

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
