# Peer review of "Predicting the Effect of Mo Addition on Metastable Phase Equilibria and Diffusion Path of Fe in NiAl Laser-Clad Coatings Using First-Principle Calculations and CALPHAD Simulations"

_processes, doi:10.3390/pr10061228_

Round 1

Reviewer 1 Report

  1. Highlight the article innovation point in the abstract. Why should anyone read your article?
  2. The main weakness of this paper is they have not provided the application of the present study in the real-world problem and are suggested to provide.
  3. Outlook and future perspectives can be included in the conclusion part.
  4. Heading should be numbered or sub-numbered for the whole manuscript.
  5. The introduction section is rewritten and modified with latest articles.
  6. The conclusion part should be specific. Authors need to restate the main arguments in brief with supporting evidence and summarize all the key points made throughout with further applications. Need to make sure that it is linked to Results and Discussion.
  7. This paper should be edited grammatically.
  8. A comparison data for validation is important.
  9. Figures are recommended with more descriptive caption and explanation.
  10. Author need to mention briefly with application of present work in real world, what is new in this model and why is it considered?
  11. Would it be possibile for the authors to compare their predictions with experiments from the literature, if available?
  12. Indicate what this paper brings new compared to what already exists.

Reviewer 2 Report

I can recommend  the paper  "Predicting the effect of Mo addition on metastable phase equilibria and diffusion path of Fe in NiAl laser-clad coatings using first-principle calculations and CALPHAD simulationsfor publication in journal . “    “Processes”

 In this paper was theoretical study  the diffusion processes in nickel films..

For studies of this systems used  the quantum modeling   analysis .

 The calculation results confirmed the existence of a two-phase NiAl matrix with a molybdenum-rich phase.

I think this paper will be interesting for readers of this journal .

I am recommending to include in the references the next publications:

1. S. P. Repetsky, I. G. Vyshyvana,Y.Nakazawa,S. P. Kruchinin, S.Bellucci., Electron transport in carbon nanotubes with adsorbed chromium impurities. Materials (2019), 12, 524.

2, S. P. Repetsky, I. G. Vyshyvana, S. P. Kruchinin, S.Bellucci., Influence of the ordering of   impurities on the appearance of an energy gap and on the electrical conductance of graphene. Scientific Reports, 8:9123(2018).

Reviewer 3 Report

The Authors described researches concerning on Predicting the effect of Mo addition on metastable phase equi- 2 libria and diffusion path of Fe in NiAl laser-clad coatings using 3 first-principle calculations and CALPHAD simulations. The manuscript could be published in Processes after major revision.  Below, several aspects  have mentioned, which should be corrected and some doubts should be explained.

  1. The Abstract should contain the most important results from the manuscript.
  2. The motivation of studies should be highlighted in the Introduction section.
  3. All reflexes should be recognized and marked in the XRD pattern.
  4. The Discussion is very poor. The Authors should compare their results to data delivered by other investigators.
  5. The Conclusions are too long. This part is summary of conducted studies.

Generally, the Authors did some excellent work. However it could not be published in present form in Processes. I recommend major revision.

Round 2

Reviewer 1 Report

Paper can be accepted in the resent form 

Reviewer 3 Report

The Authors improved the manuscipt. It could be published in present version.